# Isolation and Characterization of Two Novel Siphoviruses Novomoskovsk and Bolokhovo, Encoding Polysaccharide Depolymerases Active against *Bacillus pumilus*

**DOI:** 10.3390/ijms232112988

**Published:** 2022-10-26

**Authors:** Anna V. Skorynina, Olga N. Koposova, Olesya A. Kazantseva, Emma G. Piligrimova, Natalya A. Ryabova, Andrey M. Shadrin

**Affiliations:** 1G.K. Skryabin Institute of Biochemistry and Physiology of Microorganisms, Pushchino Scientific Center for Biological Research of the Russian Academy of Sciences, Federal Research Center, Prospect Nauki, 5, 142290 Pushchino, Russia; 2Institute of Protein Research RAS, Institutskaya st., 4, 142290 Pushchino, Russia

**Keywords:** bacteriophage, *Bacillus pumilus*, *Andromedavirus*, depolymerase, biocontrol, genome sequencing

## Abstract

This study describes two novel bacteriophages infecting members of the *Bacillus pumilus* group. Even though members of the group are not recognized as pathogenic, several strains belonging to the group have been reported to cause infectious diseases in plants, animals and humans. *Bacillus pumilus* group species are highly resistant to ultraviolet radiation and capable of forming biofilms, which complicates their eradication. Bacteriophages Novomoskovsk and Bolokhovo were isolated from soil samples. Genome sequencing and phylogenetic analysis revealed that the phages represent two new species of the genus *Andromedavirus* (class *Caudoviricetes*). The phages remained stable in a wide range of temperatures and pH values. A host range test showed that the phages specifically infect various strains of *B. pumilus*. The phages form clear plaques surrounded by halos. Both phages Novomoskovsk and Bolokhovo encode proteins with pectin lyase domains—Putative depolymerases. Obtained in a purified recombinant form, the proteins produced lysis zones on the lawn of a *B. pumilus* strain. This suggests that Novomoskovsk and Bolokhovo may be effective for the eradication of *B. pumilus* biofilms.

## 1. Introduction

Bacilli are ubiquitous bacteria that occupy different ecological niches: soil, fresh and seawater, and can be transferred with air. They are highly resistant to unfavorable factors due to their ability to form spores. The *Bacillus pumilus* group of bacilli, also known as *B. pumilus sensu lato* (*s.l.*), are the most resistant to extreme conditions such as H_2_O_2_-mediated oxidative stress [1], high salt concentration [2], UV radiation [1,3], high-speed accelerations [3] and ionizing radiation [4]. As of 2022, the group includes six species: *B. pumilus sensu stricto* (*s.s.*), *B. safensis* [5], *B. altitudinis* [6], *B. xiamenensis* [7], *B. zhangzhouensis* and *B. australimaris* [8].

*B. pumilus s.l*. spores can remain viable under harsh conditions simulating the surface of Mars [9]. *B. safensis* and *B. pumilus s.s.* were found at the assembly site of the Phoenix spacecraft [5,10], which was used to explore the Mars surface in 2008, suggesting that the planet may now be contaminated with these bacteria. Because of their high viability, *B. pumilus s.l.* are widely used as reference microorganisms when determining effective radiation doses for sterilization purposes [11]. A large number of *B. pumilus s.l.* strains are known to produce industrially valuable enzymes [12,13,14,15,16] and metabolites [17]. It was also shown that they can exert a probiotic effect [18,19], suppressing the growth of pathogenic microflora in the rhizosphere [20,21] and thereby improving the fitness of certain plants. On the other hand, several cases have been described of *B. pumilus s.s.* strains causing food poisoning [22,23]. The cytotoxic properties of such strains are due to the synthesis of cyclic lipopeptides [22] and/or proteins with hemolytic activity, neither of which is typical of *B. pumilus s.s*. [23].

*B. pumilus s.s.*, like many other bacilli, is capable of forming biofilms on various surfaces. The formation of such exopolysaccharide structures is a defense mechanism protecting bacteria from biological, physical and chemical influences. Biofilms adversely affect industrial processes [24,25,26] and are associated with a wide range of problems in medicine [27]. Biofilm formation, along with the ability to form spores, greatly complicates *B. pumilus s.s.* control and eradication.

Bacteriophages and their lytic enzymes are considered promising antibacterial agents [28,29,30,31,32]. Many studies demonstrate the ability of phages and their proteins to eradicate, partially or completely, bacterial biofilms and spores [33,34,35,36,37]. Phages and their components are also notable for their higher selectivity compared to conventional antibacterial agents, and therefore can be effectively applied in medicine where the goal is to control particular bacterial strains or species without affecting the others. Recently, a novel bacteriolytic enzyme from a *B. pumilus* bacteriophage has been characterized—Bacterial cell wall hydrolase Ply67 [36]. The enzyme has a number of outstanding properties such as sporolytic activity, stability in a broad range of temperatures and pH values, and broad specificity and activity against spores of the *B. cereus* group species [36].

Thus, it may be that *B. pumilus s.l*. bacteriophages are underestimated as a source of new bacteriolytic enzymes. In the present study we describe two novel bacteriophages infecting *B. pumilus s.l.*, Novomoskovsk and Bolokhovo, which represent two new species of the genus *Andromedavirus* (class *Caudoviricetes*) and exhibit depolymerase activity. There are currently no studies describing in vitro properties of depolymerases of the *Andromedavirus* phages. We believe that the polysaccharide depolymerases of Novomoskovsk, Bolokhovo and related phages should be subject to thorough investigation, as they are a promising tool for controlling contamination caused by the *B. pumilus* group species.

The phages were isolated from soil samples collected in the Tula region, Russian Federation, and their physiological characteristics, host specificity, genomic and morphological organization were determined. The genomes of both phages were sequenced, and their putative depolymerase genes were cloned and expressed in *E. coli*. The recombinant depolymerases demonstrated lytic activity against *B. pumilus*.

## 2. Results and Discussion

### 2.1. Phage Isolation, Plaque Morphology and Transmission Electron Microscopy

The Bolokhovo and Novomoskovsk phages were isolated from soil samples. The phages were named after the eponymous towns in the Tula region, Russian Federation, where the soil samples were collected. On the lawn of the sensitive bacterial strain *B. pumilus* AVS-01, both phages formed large transparent plaques surrounded by turbid halos (Figure 1a). The halos expanded significantly over time and probably appeared due to the presence of extracellular depolymerases.

The purified phages were studied using transmission electron microscopy (TEM). TEM has shown that Novomoskovsk has a long noncontractile tail, approximately 148 ± 8 nm in length, and a nonelongated icosahedral capsid approximately 65 ± 4 nm in diameter (Figure 1b), the two features typical of the siphovirus B1 morphotype [38].

### 2.2. Genome Characteristics

The chromosomes of Bolokhovo and Novomoskovsk are linear double-stranded DNA molecules with GC-content of 42%. The genome lengths are 49,683 bp and 49,258 bp, respectively. A whole-genome pairwise BLASTN alignment revealed that the two phages are closely related (identity 81.4%, coverage 86%). Each genome contains 81 open reading frames (ORFs). Genome annotations are presented in Appendix A.

The number of functionally assigned genes was 48 (59%) for Bolokhovo and 43 (53%) for Novomoskovsk. The phages have highly similar genomic organization: they share 68 proteins, as determined using CoreGenes 5.0. All identified ORFs were classified into seven functional groups: (i) Structural and packaging genes; (ii) host cell lysis; (iii) replication and recombination; (iv) transcriptional regulation; (v) proteins containing signal sequences; (vi) proteins with other functions; and (vii) proteins with unknown functions (Figure 2 and Figure 3).

In both genomes, most numerous genes are those encoding structural and packaging proteins. Interestingly, some of the tail genes of the two phages were predicted to encode proteins containing domains typical of pectin/pectate lyases. Additionally, three proteins participating directly in host cell lysis were identified: two holins and one N-acetylmuramoyl-L-alanine amidase. DNA replication and recombination proteins of the two phages include: DNA helicase, primase, DNA polymerase, Holliday junction resolvase and RecB-family exonuclease.

Both phages encode several proteins associated with plasmid replication and segregation: TrbC/VIRB2 protein [39], Replic_Relax superfamily protein, FtsK/SpoIIIE family protein [40] and a small DNA-binding protein containing a ribbon–helix–helix (RHH) domain similar to that of the members of the MetJ/Arc-superfamily. RHH domain-containing DNA-binding proteins are the components of plasmid segregation systems found in plasmid prophages. Such systems, when functional, always include three components: a motor protein (ParM, ParA or TubZ), a DNA-binding adapter protein (of the MetJ/Arc-superfamily or their functional homologs) and a centromere-like DNA region [41]. Additional proteins facilitating plasmid replication, such as Replic_Relax and FtsK/SpoIIIE family proteins, are often encoded next to the plasmid segregation systems. No genes of the motor proteins were identified in the Novomoskovsk and Bolokhovo genomes, which may indicate that the phages were once able to form circular plasmid prophages, but lost that ability in the course of evolution.

Ninety percent of the genes in both genomes are located on the same DNA strand. No RNA-polymerase genes were found; however, the phages possess RNA-polymerase sigma70-family sigma factors: gp44 (Bolokhovo) and gp48 (Novomoskovsk). They also encode putative transcription factors (gp40 and gp43 in the Bolokhovo and Novomoskovsk genomes, respectively) sharing similarities with region 4 of sigma70-family sigma-factors, which, in a complex with the β-flap domain of RNA-polymerase, is responsible for recognition of the −35 promoter element sequence [42]. This suggests that the phages modulate bacterial RNA-polymerase upon infection. No genes of DNA-binding transcription factors were found. Usually, such transcription factors share similarity with the CI, CII and CIII proteins of phage lambda, the homologs of which control the lysis-lysogeny switch in many temperate phages [43,44,45]. The absence of DNA-binding transcription factors may indicate the virulent lifestyle of Novomoskovsk and Bolokhovo—a conclusion also supported by the observed transparent phenotype of the phage plaques, as well as by the *B. pumilus* AVS-01 growth kinetics upon the infections with the phages.

### 2.3. Comparative Genomics

The tBLASTx whole-genome alignment of Bolokhovo and Novomoskovsk (Figure 4a) illustrates similar genomic organization of the two phages. The region with the lowest identity (in the vicinity of the 20 kb marker in Figure 4a) corresponds to the genes of the pectinlyase-like superfamily proteins.

In the proteomic tree generated using ViPTree (partially shown in Figure 4b), Bolokhovo and Novomoskovsk are found on different branches of the same clade comprised solely of the members of the genus *Andromedavirus* (class *Caudoviricetes*). *Bacillus* phages Finn and Riggi are the closest to Novomoskovsk, with a nucleotide identity of 83.4% and 81.4%, respectively, as determined with EMBOSS Stretcher. In the case of Bolokhovo, the closest relative is *Bacillus* phage Curly (92.4% nucleotide identity). With the current criterion for viral species demarcation being <95% nucleotide identity [46], Novomoskovsk and Bolokhovo can be classified as separate species of the genus *Andromedavirus*.

### 2.4. Time-Kill Assay of Bolokhovo and Novomoskovk Phages

One characteristic essential to the practical application of bacteriophages is phage lytic activity, which enables determination of the effective phage concentration as well as lysis time adjustment.

To assess the lytic activity of the studied phages, a *B. pumilus* AVS-01 culture was infected at MOI of 0.1, 1 and 10 and incubated for 150 min until lysis was observed.

As shown in Figure 5, the lytic activity of Novomoskovsk and Bolokhovo grows as MOI increases. Both Bolokhovo and Novomoskovsk were capable of completely inhibiting the growth of *B. pumilus* AVS-01 within 2 h of infection, even at MOI of 0.1, which indicates a high killing efficiency of the studied phages.

### 2.5. Adsorption Assay and One-Step Growth Curves

The adsorption dynamics of the two phages were slightly different: the adsorption of Bolokhovo was delayed at the initial stage, as compared to Novomoskovsk (Figure 6a).

After 8 min, however, the percentage of adsorbed virions reached about 80% in both phages. The adsorption endpoints and the sizes of the residual fraction of the two phages were found to be similar, as were the adsorption constants, which measured 1.84 ± 0.27 × 10^−8^ mL/min for Novomoskovsk and 1.72 ± 0.07 × 10^−8^ mL/min for Bolokhovo.

The burst sizes of Bolokhovo and Novomoskovsk were estimated to be 80 ± 21 PFU and 113 ± 47 PFU per infected cell, respectively (Figure 6b). The phages had a similar latent period of about 20 min, which is consistent with the synchronicity of their lytic curves (Figure 5). The rise period was found to be 15 min for Bolokhovo and 20 for Novomoskovsk; the difference may stem from a longer maturation time of Novomoskovsk, which is consistent with a larger burst size.

### 2.6. Thermal and pH Stability of Phages

Phage stability was assessed at different temperatures and pH values. Both phages remained viable in a broad range of temperatures and pH values. Bolokhovo remained stable (10^9^ PFU/mL) at pH values ranging from 4 to 11, and at temperatures ranging from 4 °C to 50 °C (Figure 7a,c); whereas at 60 °C, the phage titer dropped significantly (10^6^ PFU/mL). Novomoskovsk remained stable (10^9^ PFU/mL) at temperatures ranging from 4 °C to 60 °C and pH values from 4 to 10 (Figure 7b,d).

### 2.7. Divergence and Domain Organisation of Novomoskovsk and Bolokhovo Depolymerases

Negative colonies of Novomoskovsk and Bolokhovo are surrounded by halos (Figure 1a). Such halos may occur because of the degradation of bacterial mucus mediated by soluble forms of phage depolymerases [47,48,49]. Due to a smaller size, the enzymes diffuse in soft agar faster than bacteriophages; therefore, they can digest bacterial capsules that have not been affected by phage endolysins or tail depolymerases [31]. Both Novomoskovsk and Bolokhovo encode depolymerases of the Pectin_lyas_fold superfamily (InterPro code: IPR012334) (gp28 and gp31 in the Bolokhovo and Novomoskovsk genomes, respectively) (Appendix A).

Gp28 contains a Pectate_lyase_SF_prot domain (IPR024535) in its N-terminal part and a Beta_helix domain (IPR039448) at the C-terminus, while Gp31 only has a Beta helix domain in its C-terminal region. Pectinlyase-like sequences and beta-helices are typical for phage depolymerases described in other studies [30,31,47,50,51]. A BLASTp alignment of the putative depolymerases of Bolokhovo and Novomoskovsk revealed that the proteins are 29.73% identical to each other, the coverage being 99%. An HHpred analysis revealed that both Gp28 and Gp31 have homologs among the domains and proteins of the Pectinlyase-like superfamily (SCOPe ID: b.80.1) [52]. For instance, HHpred probability values of 99% were obtained in the alignments of both proteins with the rhamnogalacturonase A of *Aspergillus aculeatus* (SCOPe ID: d1rmga_), suggesting that Gp28 and Gp31, though highly divergent in sequence, are structurally and functionally homologous.

The Maximum Likelihood phylogram inferred from the multiple alignment of Gp28, Gp31 and different endolysins and depolymerases of the members of *Andromedavirus* is shown in Figure 8. Significantly diverged from one another, Gp28 and Gp31 fall into different clades. The two proteins being so divergent, it is reasonable to expect them to differ considerably in their key characteristics, such as affinity to bacterial glycocalyx.

### 2.8. Cloning and Expression of Capsule Depolymerases

The depolymerase genes were cloned and the recombinant proteins were purified from *E. coli* as described in Materials and Methods. The electrophoretic mobility of the purified proteins in polyacrylamide gel was similar to that of the 60 kDa marker, which is consistent with the calculated molecular weight of the monomers (59.4 kDa—Gp28, 58.5 kDa—Gp31). The purity of the depolymerases in the preparations was at least 85% for Gp28 (Figure 9a) and 70% for Gp31 (Figure 9b).

The ability of the depolymerases to form zones of lysis on the lawn of a *B. pumilus* strain was assessed in a spot test (Figure 9c,d). The depolymerase of Bolokhovo (Gp28) was highly effective: The lytic zones produced by the enzyme were seen clearly, even when only 5 ng of the depolymerase preparation was spotted (Figure 9c).

The oligomeric state of the depolymerases was determined by size exclusion chromatography on a prepacked HiPrep^TM^ 16/60 Sephacryl^TM^ S-200 HR column (GE Healthcare). For both Gp28 and Gp31 preparations, fractions exhibiting depolymerase activity were eluted with approximately the same elution volume as human IgG, suggesting molecular weights of ~160 kDa. These molecular weights, determined under native conditions, are consistent with the homotrimeric structure of the Pectinlyase-like superfamily proteins.

### 2.9. Determination of Strain Specificity of Phages and Depolymerases

Out of the 12 *B. pumilus* strains tested, Bolokhovo and Novomoskovsk were able to infect 9 and 11, respectively. Neither phage exhibited an ability to form plaques on the other 10 strains tested: 9 of the genus *Bacillus* and 1 *E. faecium* (Table 1).

The lytic activity of the depolymerases was analyzed on the same 12 strains of the *B. pumilus* group. The strain specificity of Gp28 corresponds to that of its source phage, Bolokhovo, which is often the case with phage depolymerases. Gp31, however, was found to have much narrower specificity compared to its source phage, Novomoskovsk, as the enzyme, was active against only 3 out of the 12 strains tested.

Of the three strains susceptible to Gp31, two were also sensitive to Gp28 (VKM B-23 and VKM B-731).

### 2.10. Evaluation of Thermal Stability and pH Stability of Depolymerases

Enzyme stability is a crucial characteristic for evaluating the range of conditions in which an enzyme can potentially be used.

To assess the stability of enzymes, we analyzed their activity 1 h after incubation at temperatures ranging from 4 to 70 °C and pH ranging from 4.0 to 11.0. The results were graded based on the presence or absence of a clearing zone on a bacterial lawn after treatment with Gp28 and Gp31 proteins. Gp28 produced clearing zones at pH ranging from 4.0 to 8.0 at concentrations of 0.1, 0.01 and 0.001 µg/µL. At pH 9.0, clearing zones were observed only at concentrations of 0.1 and 0.01 µg/µL, and at pH 10.0 and 11.0. Only at 0.1 µg/µL. Gp31 produced clearing zones at pH from 4.0 to 11.0, but only at a concentration of 0.1 µg/µL.

Both depolymerases remained active at a pH ranging from 4.0 to 11.0. At pH 12.0, both enzymes lost their activity, in contrast to their source phages, Novomoskovsk and Bolokhovo. At the same time, the range of thermal stability of Gp28 and Gp31 was found to be wider than that of the source phages: 4–70 °C as opposed to 4–60 °C in the phages.

## 3. Materials and Methods

### 3.1. Bacterial Strains and Cultivation Conditions

In this study, we used bacterial strains obtained from different sources (Table 1). Lysogeny broth (LB), LB agar (1.5% *w*/*v* and 0.75% *w*/*v*) and Ca-Mg-enriched LB media (LB with additional 10 mM CaCl_2_ and 10 mM MgCl_2_) were used for bacterial cultivation and phage propagation. All cultures were grown at 37 °C, unless stated otherwise.

### 3.2. Isolation and Propagation of Bacteriophages

Phages Novomoskovsk and Bolokhovo were isolated from soil samples. For each phage, 1 g of the soil sample was added to a *B. pumilus* AVS-01 culture at the optical density (OD590) of 0.2 and the mixture was incubated for 3 h at 37 °C [55]. The mixture was then centrifuged at 10,000× *g*, 4 °C for 10 min to remove cell debris, followed by filtration through a 0.22-µm sterile filter. The filtrates were tested for the presence of bacteriophages by the double layer agar technique [56], using the sensitive host *B. pumilus* AVS-01. For phage extraction and purification, negative colonies were separately transferred into 1.5-mL polypropylene test tubes with 1 mL of SM buffer [50 mM Tris pH 7.5; 100 mM NaCl; 1 mM MgSO_4_; 10 mM CaCl_2_; 10 mM MgCl_2_; 0.01% gelatin] and incubated with shaking for 4 h at 4 °C. The obtained extracts were titrated by serial dilutions on the lawn of the sensitive host *B. pumilus* AVS-01. In order to exclude the presence of other phages with morphologically similar plaques, the extraction–titration cycles were repeated three times.

In order to obtain high-titer preparations, phage propagation and polyethylene glycol (PEG) precipitation were performed as described in our previous work [57], with modifications regarding the propagation host (*B. pumilus* AVS-01) and incubation temperature (37 °C). The obtained high-titer preparations were purified by CsCl density gradient centrifugation. For this purpose, 3 mL of each preparation was loaded on a pre-formed CsCl gradient (1.3; 1.4; 1.5; 1.6; 1.7 g/mL) and centrifuged at 30,000 rpm for 2.5 h.

### 3.3. Transmission Electron Microscopy

CsCl-purified phage suspensions were applied onto 400-mesh carbon formvar-coated copper grids and stained with uranyl acetate (1% *m*/*v*). JEM-100C transmission electron microscope (JEOL, Tokyo, Japan) was used for visualization at an accelerating voltage of 80 kV. Images were recorded on Kodak SO-163 film (Kodak, Cat. No. 74144, Hatfield, PA, USA) at a magnification of 45,000×. Phage particle dimensions were measured for 10 particles using ImageJ version 1.53e [58] in relation to the scale bar generated by the microscope software.

### 3.4. Isolation of Bacteriophage DNA

In order to remove bacterial nucleic acids, DNAse I reaction buffer with MgCl_2_ (Thermo Scientific, Waltham, MA, USA), DNAse I (Thermo Scientific, Waltham, MA, USA; 10 U/mL), and RNAse A (Sigma-Aldrich, Saint Louis, MO, USA; 50 U/mL) were added to phage preparations, followed by incubation for 2 h at 37 °C. Phage genomic DNA was isolated by phenol–chloroform extraction and isopropanol precipitation [59], dissolved in deionized water, and stored at −20 °C.

### 3.5. Genome Sequencing, Annotation and Comparative Genomics

Phage DNA was sequenced on the Illumina platform, using the TruSeq library preparation protocol. Phage genome sequences were assembled using SPAdes v. 3.11.1 [60]; open reading frames (ORF) were identified using RASTtk [61]. Gene annotation was performed with BLAST (NCBI) [62] and HHpred [63]. Signal peptides were found using SignalP 5.0 [64]. The circular genome maps were visualized using BLAST Ring Image Generator (BRIG) [65]. CoreGenes 5.0 [66] was used to calculate shared protein content. The phylogenetic proteomic tree was constructed using the ViPTree server version 1.9 [67]. Phage whole-genome nucleotide alignment was performed using EMBOSS Stretcher [68].

### 3.6. Killing Assay of Bolokhovo and Novomoskovk Phages

In order to determine the lytic activity of the Novomoskovsk and Bolokhovo phages, the *B. pumilus* AVS-01 strain was grown in Ca-Mg-enriched LB medium to OD590 of 0.2 (1 × 10^7^ PFU/mL). Then, 180-µL aliquots of the bacterial culture were transferred to a 96-well microplate and mixed with 20 µL of phage preparations in SM buffer (1 × 10^7^, 1 × 10^8^, 1 × 10^9^ PFU/mL) to ensure MOI of 0.1, 1 and 10, respectively. The same bacterial culture mixed with 20 µL of SM buffer was used as a control. Incubation was carried out for 150 min at 37 °C in a FilterMax F5 microplate reader (Molecular Devices) with OD590 being measured every 10 min. Five trials of the experiment were carried out for each phage, and the data were visualized in GraphPad Prism 8.4.3. as the mean ± standard deviation.

### 3.7. Adsorption Assay

Adsorption analysis was performed as described previously [69], according to the protocol originally developed by Kropinski [70], with some modifications. The experiment was carried out in 100-mL flasks in a water bath at a temperature of 37 °C and constant orbital shaking at 60 rpm. Nine milliliters of a mid-log phase *B. pumilus* AVS-01 culture grown in the Ca-Mg-enriched LB media to an OD590 of 0.2–0.3 (approx. 1 × 10^7^ CFU/mL) was transferred to a 100-mL flask and mixed with one milliliter of a phage preparation (1 × 10^7^ PFU/mL) to ensure a MOI value of 0.1. In the control sample, nine milliliters of non-inoculated Ca-Mg-enriched LB medium was used instead of the AVS-01 culture. An aliquot of 25 µL was obtained from the tested sample at a time point zero and then every minute over the next 13 min (altogether 14 aliquots). Three aliquots were removed from the control flask at the time points of 0, 10 and 13 min.

Each aliquot was transferred into an ice-precooled Eppendorf tube containing 940 µL of LB and 25 µL of chloroform, vortexed thoroughly, and 10 µL of the resultant mixture was tested for the presence of non-adsorbed phage particles using the two-layer agar method. For the convenience in counting plaque-forming units, incubation was carried out at 20–22 °C for 18–20 h.

The number of non-adsorbed phages in each tested sample was expressed as a percentage of the control. Five replicates of the experiment were carried out for each phage, and the data were visualized in GraphPad Prism 8.4.3. as the mean ± standard deviation.

### 3.8. One-Step Growth Curve

One-step growth curve experiment was performed as described by Hyman and Abedon [71]. Nine milliliters of a *B. pumilus* AVS-01 culture grown in Ca-Mg-enriched LB media to an OD590 of 0.2 was mixed with one milliliter of a phage preparation (1 × 10^7^ PFU/mL) to ensure a MOI value of 0.1. The mixture was incubated for eight minutes until 80% adsorption was reached (Figure 6a). Then, a 1-mL aliquot was collected and centrifuged twice at 3500× *g*, 5 °C for 2 min, the sediment resuspended each time in 1 mL of fresh medium to remove non-adsorbed phage particles. Then, 0.1 mL of the resultant suspension was transferred to 9.9 mL of LB in a 100-mL flask and incubated in a water bath at 37 °C with constant orbital shaking at 60 rpm for 40 min. During the incubation, 100-µL aliquots were removed every 2.5–5 min, which were immediately titrated in chloroform-supplemented LB medium and plated using the double-layer agar technique. The plates were incubated for 18–20 h at 20–22 °C, followed by plaque enumeration. The experiment was performed in five replicates for each phage, and the obtained data were visualized in GraphPad Prism 8.4.3 as the mean ± standard deviation.

Burst size was calculated using the formula: (A − B)/C, where:

A—Free phage, average of the time points on the plateau after the burst (samples collected between 20 and 40 min since the beginning of cultivation for the Bolokhovo phage, and between 25 and 40 min for the Novomoskovsk phage)

B—Free phage, average of the time points on the plateau before the burst (samples collected at time points 0 and 5 min)

(A − B)—Total burst (released phages)

C—Infecting phage (virions adsorbed within 8 min since the beginning of cultivation)

The burst size of each phage was calculated independently in each experimental replicate and expressed as the mean ± standard deviation.

The latent period was calculated by adding together the adsorption time, the duration of the two centrifugation-resuspension cycles, and the length of plateau before the burst.

### 3.9. Thermal and pH Stability of Phages

For a pH stability test, 20 µL of each phage suspension (10^10^ PFU/mL) was added into a test tube containing 180 µL of one of the following buffer solutions: glycine-HCl buffer (for pH 2.2 and 3.0); sodium acetate buffer (pH 4.0 and 5.0); sodium phosphate buffer (pH 6.0, 7.0 and 8.0); glycine-NaOH buffer (pH 9.0 and 10.0); or Na_2_HPO_4_-NaOH buffer (pH 11.0 and 12.0). The samples were incubated at 20 °C for 60 min, and the survival of phages was assessed in a spot test with 10-fold serial dilutions. For a thermal stability test, 40 µL of each phage suspension (10^9^ PFU/mL) was incubated at 4 °C, 20 °C, 30 °C, 40 °C, 50 °C, 60 °C, 70 °C, 80 °C and 90 °C for 60 min, and the surviving phages were enumerated in a spot test with 10-fold serial dilutions. Both tests were performed in triplicate. The obtained PFU/mL values were expressed as the mean ± standard deviation. The diagrams were created using Graph Pad Prism 8.4.3.

### 3.10. Divergence and Domain Organisation of Novomoskovsk and Bolokhovo Depolymerases

Pairwise alignments of individual proteins were performed using BLASTp [72]. InterPro [73] was used for protein classification down to the family level and for domain detection. Protein phylograms were generated in MEGAX [74], applying MUSCLE [75] for sequence alignment and the Maximum Likelihood (ML) method with the Whelan and Goldman substitution model [53] for phylogenetic inference (500 bootstrap replicates). FigTree v1.4.4 was used for visualization of the phylograms [76].

### 3.11. Cloning and Purification of Depolymerases

The genes annotated as putative depolymerases, gp28 (Bolokhovo) and gp31 (Novomoskovsk), were cloned into the pET33 expression vector at the NcoI and NotI sites with a C-terminal 6xHis-tag, using the following pairs of primers: 5′-ATATA***CCATGG***GTAACGCTACTAGATCA-3′ plus 5′-TATAT***GCGGCCGCG***ATGTTGTTTGAAGAAGCT-3′ for gp28, and 5′-ATATA***CCATGG***GAGCAAAAACAAGTAATTATG-3′ plus 5′-TATAT***GCGGCCGC***TAGGTTACCAGAAGATACA-3′ for gp31. In order to confirm the correctness of cloning, the obtained plasmids were Sanger-sequenced with the primers: 5′-TAATACGACTCACTATAGGG-3′, 5′-ATGCTAGTTATTGCTCAG-3′, 5′-CAGTGGGTACAACGGAGA-3′, 5′-CTTCTGTAGTATTTCCATTGAGA-3′ for gp28, and 5′-TAATACGACTCACTATAGGG-3′, 5′-ATGCTAGTTATTGCTCAG-3′, 5′-TATACAGGTTATAACGGTCAC-3′, 5′-ATAGACCCAAACTCGCAA-3′ for gp31.

The *E. coli* BL21(DE) Star cells were transformed with the constructed plasmids. Overnight cultures of the recombinant strains were added at a ratio of 1:100 to fresh LB media and grown at 37 °C until an OD600 of 0.4–0.6 was reached. Then, 0.1 mM IPTG was added and cultivation continued overnight at 25 °C. The cultures were centrifuged for 15 min at 3000 rpm, 4 °C, and the pellets were resuspended in Buffer A [40 mM Tris-HCl pH 7.5; 0.5 M NaCl; 5% glycerol] and sonicated on ice (amplitude 50%, 3 cycles of 30 s each), using a Q700-sonicator with ½ inch probe. The histidine-tagged proteins were purified from the lysate by metal-chelate chromatography on a HiTrap column (GE Healthcare) according to the manufacturer’s instructions, and dialyzed against Buffer D [20 mM Tris-HCl pH 7.5; 150 mM NaCl]. Purification quality was assessed by SDS gel electrophoresis in 15% polyacrylamide gel (sdsPAGE).

### 3.12. Phage Host Range

A host range test was performed using 12 *B. pumilus* strains, two *B. cereus* strains, two *B. thuringiensis* strains, one *B. weihenstephanensis* strain, one *B. flexus* strain, two *B. subtilis* strains, one *B. megaterium* strain and one *Enterococcus faecium* strain. The test was performed as described previously [57] with differences in incubation conditions (37 °C, 24 h). The results were obtained in three replicates for each strain.

### 3.13. Strain Specificity of Depolymerases

The spectrum of *B. pumilus* strains sensitive to the depolymerases was determined by a spot test using 12 strains from our laboratory collection. Briefly, 100 µL of an overnight culture of each strain was mixed with 3.5 mL of 0.5% LB-agar and overlaid onto 1.5% LB-agar in a Petri dish, and the covered dish was left to air-dry for an hour at room temperature. Then, three microliters of purified depolymerase solutions (0.1 μg/µL in buffer D) were dripped onto the agar plate in the Petri dish. Buffer D was used as a control. The plates were incubated at 30 °C for 9 h. The analysis was carried out in four biological replicates.

### 3.14. Evaluation of Thermal Stability and pH Stability of Depolymerases

The thermal stability of the depolymerases was evaluated at the temperatures of 4 °C, 20 °C, 30 °C, 40 °C, 50 °C, 60 °C, 70 °C, 80 °C and 90 °C. Twenty-five microliters of each depolymerase solution (0.1 μg/μL in Buffer D) was incubated at a given temperature for an hour. Buffer D was used as a control. For a convenient result interpretation, the solutions were additionally diluted prior to the spot test to final concentrations of 0.1, 0.01 and 0.001 µg/µL. The remaining enzyme activity was evaluated in a spot test on the lawns of the two *B. pumilus* strains that were found to be most susceptible to the enzymes in the strain specificity analysis: VKM B-23 (for Gp28) and VKM B-749 (for Gp31). The lawns were prepared in the same way as in the strain specificity experiment (Section 3.13). Plates were incubated in the same conditions as in the phage thermal stability experiment (at 30 °C for 9h), so as to facilitate careful comparison of the phages and their enzymes. The assay was carried out in three biological replicates.

For a pH stability assay, the depolymerases were diluted in the following buffers to the concentration of 0.1 µg/µL: glycine-HCl buffer (for pH 2.2 and 3.0); sodium acetate buffer (pH 4.0 and 5.0); sodium phosphate buffer (pH 6.0, 7.0 and 8.0); glycine-NaOH buffer (pH 9.0 and 10.0); and Na_2_HPO_4_-NaOH buffer (pH 11.0 and 12.0). The solutions were incubated at 37 °C for 1h. The same buffers were used as controls. The remaining enzyme activity was evaluated in a spot test as described in the paragraph above. Prior to the spot test, the solutions were additionally diluted in the corresponding buffers to final concentrations of 0.1, 0.01 and 0.001 µg/µL. Three microliters of the solutions used per spot.

## 4. Conclusions

In this study, two novel bacteriophages of *B. pumilus sensu lato*, Novomoskovsk and Bolokhovo were isolated and characterized. Based on their morphology, nucleotide identity and whole-genome phylogenetic analysis, the phages can be classified into two new species of the genus *Andromedavirus* (class *Caudoviricetes*). Novomoskovsk and Bolokhovo were able to infect most of the *B. pumilus* strains tested; however, they could not infect nine other *Bacillus* strains and one strain of *E. faecium*. The phages proved stable during 1-h incubation in a wide range of temperatures and pH values. Both bacteriophages formed clear plaques surrounded by turbid halos due to the presence of depolymerases belonging to the Pectin_lyas_fold superfamily. Homologs of these enzymes, according to a BLASTP analysis, are found only in phages belonging to the genus *Andromedavirus*.

The isolated and purified depolymerases Gp28 and Gp31 showed high stability in temperatures ranging from 4 to 70 °C and pH from 4.0 to 11.0. The enzymes have a low BlastP identity to each other, and a different range of susceptible strains, suggesting that they have different substrate specificity, which makes their combined use promising.

## Figures and Tables

**Figure 1 ijms-23-12988-f001:**
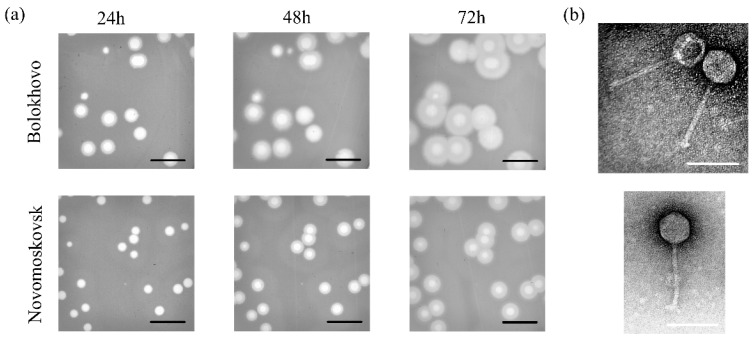
Plaque morphology and transmission electron microscopy. (**a**) Phage plaques on the lawn of *B. pumilus* AVS-01. Scale bar: 1 cm. (**b**) Transmission electron microscopy of bacteriophage particles negatively stained with 1% (*w*/*v*) uranyl acetate. Scale bar: 100 nm.

**Figure 2 ijms-23-12988-f002:**
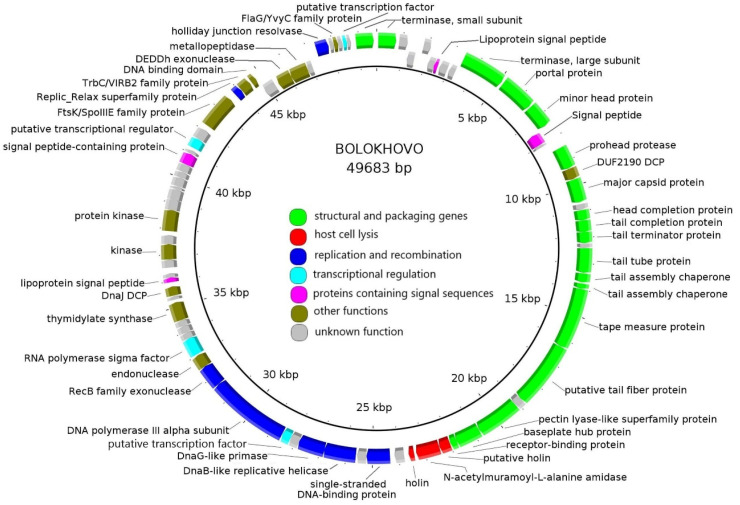
The Bolokhovo genome map. Functionally assigned ORFs are colored according to their general functions (see the color labeling scheme in the center).

**Figure 3 ijms-23-12988-f003:**
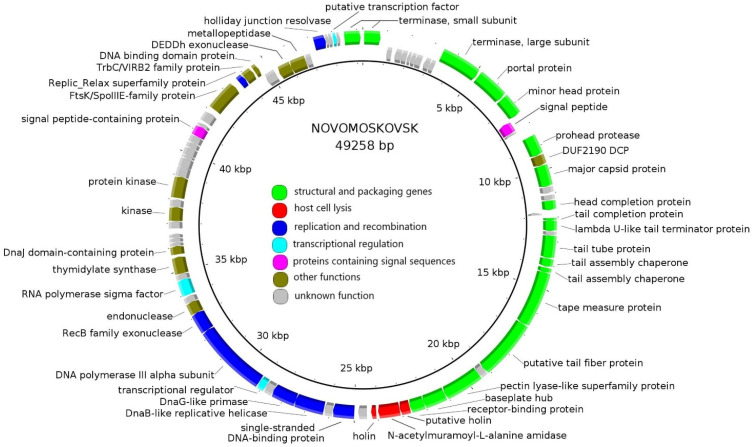
The Novomoskovsk genome map. Functionally assigned ORFs are colored according to their general functions (see the color labeling scheme in the center).

**Figure 4 ijms-23-12988-f004:**
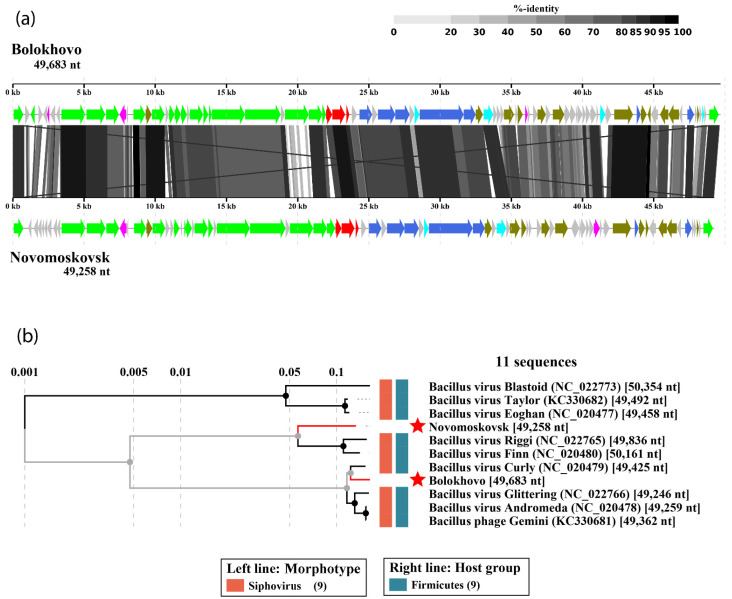
The VipTree analysis of Novomoskovsk and Bolokhovo. (**a**) The TBLASTX comparison of the two genomes. The gene-coloring scheme is identical to that in Figure 2 and Figure 3. The gray regions between the genome maps indicate the level of identity, from low (light-gray) to high (black, see the coloring scheme on the top). (**b**) Part of the proteomic tree containing Novomoskovsk, Bolokhovo and other members of the genus *Andromedavirus*.

**Figure 5 ijms-23-12988-f005:**
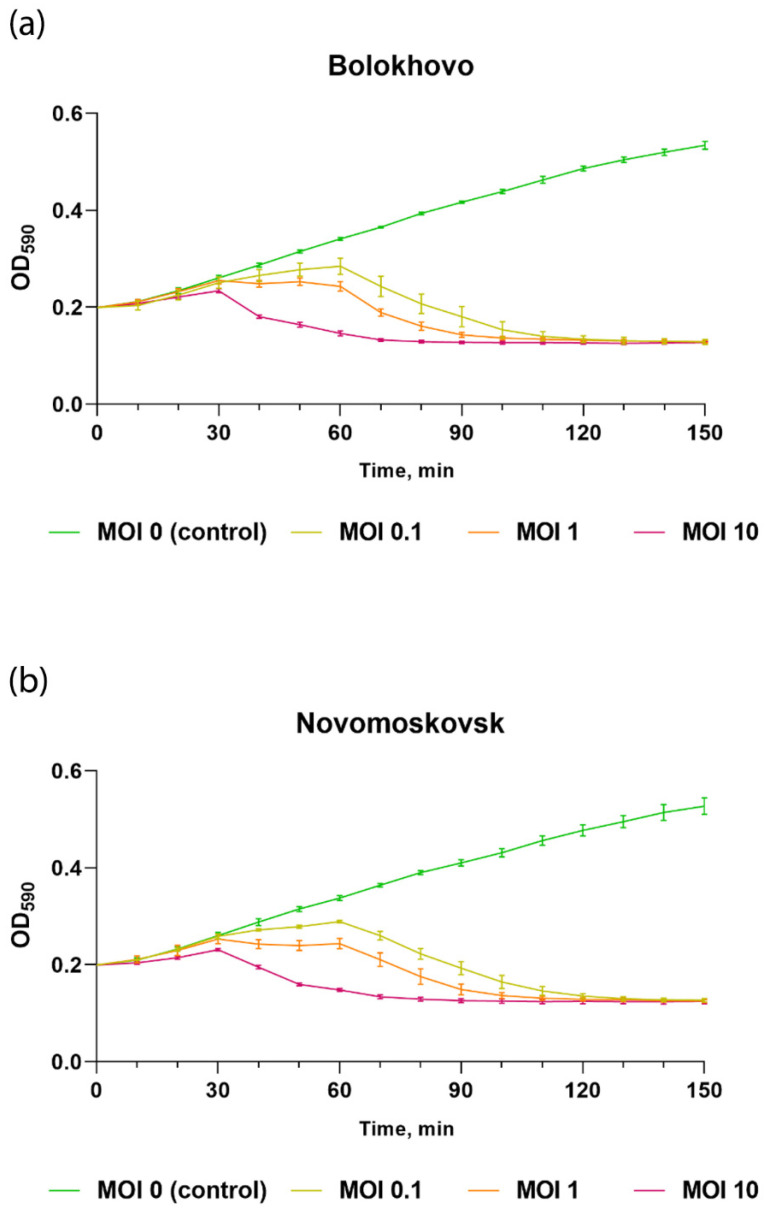
Time-kill assay of Bolokhovo and Novomoskovsk phages. *B. pumilus* AVS-01 growth kinetics upon infection with (**a**) Bolokhovo and (**b**) Novomoskovsk at different MOI. The error bars represent the standard deviation from the means of the three independent trials.

**Figure 6 ijms-23-12988-f006:**
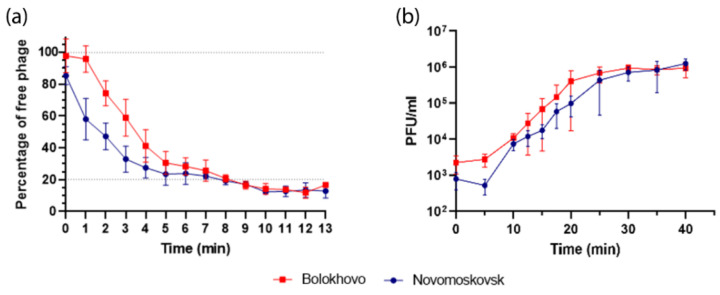
Growth dynamics of Novomoskovsk and Bolokhovo on *B. pumilus* AVS-01. (**a**) Adsorption assay, (**b**) one-step growth curve assay.

**Figure 7 ijms-23-12988-f007:**
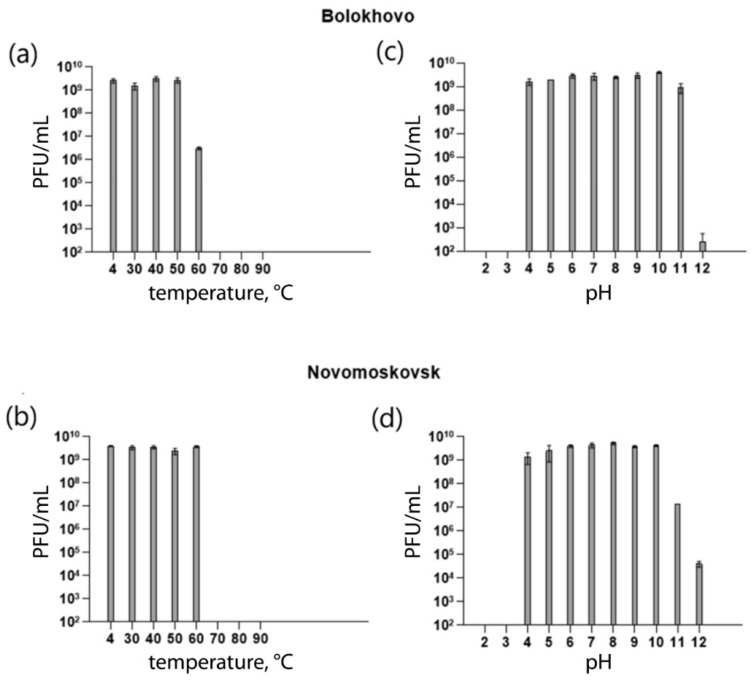
Thermal and pH stability of the Bolokhovo and Novomoskovsk phages. (**a**,**b**) Thermal stability diagrams; (**c**,**d**) pH stability diagrams. The results are expressed as the mean ± standard deviation for three replicates.

**Figure 8 ijms-23-12988-f008:**
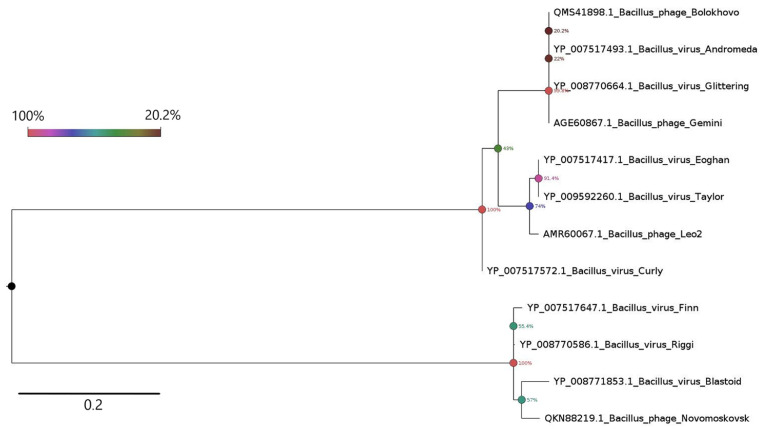
Phylogenetic analysis of the depolymerases of Novomoskovsk, Bolokhovo and the members of *Andromedavirus*. The phylograms were constructed using the Maximum Likelihood method [53] with 500 bootstrap replicates. The scale bars represent the number of amino acid substitutions per site.

**Figure 9 ijms-23-12988-f009:**
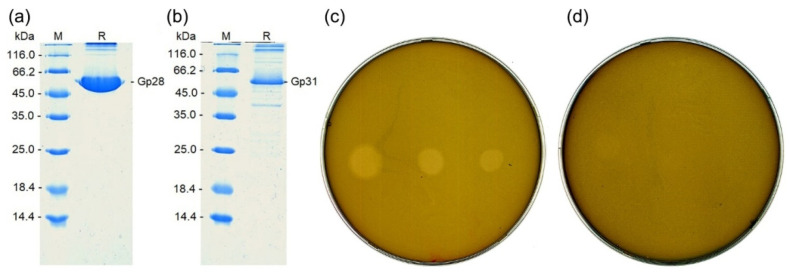
The properties of Bolokhovo (Gp28) and Novomoskovsk (Gp31) recombinant depolymerases. (**a**,**b)** The composition of Gp28 and Gp31 preparations determined by sdsPAGE. Track M—Molecular weight markers (Thermo Scientific, No. 26610). (**c**,**d)** Spot tests of Gp28 and Gp31 (respectively) on *B. pumilus* VKM B-23. On each plate, enzyme concentrations were: Left spot—500 ng, middle spot—50 ng, right spot—5 ng. Five microliters of Buffer D were used as control samples (spots are unobservable).

**Table 1 ijms-23-12988-t001:** The host range of the Bolokhovo and Novomoskovsk bacteriophages and the strain specificity of Gp28 и Gp31.

No.	Organism	Strain	Source	Bolokhovo		Novomoskovsk	
				Phage Lysis	Gp28 Lysis	Phage Lysis	Gp31 Lysis
1	*B. pumilus s.l.*	AVS-01	Laboratory collection	+	+	+	−
2	*B. pumilus*	VKM B-23	VKM	+	+	+	+
3	*B. pumilus*	VKM B-93	VKM	+	+	+	−
4	*B. pumilus*	VKM B-423	VKM	+	+	+	−
5	*B. pumilus*	VKM B-424	VKM	+	+	+	−
6	*B. pumilus*	VKM B-428	VKM	−	−	−	−
7	*B. pumilus*	VKM B-508	VKM	+	+	+	−
8	*B. pumilus*	VKM B-731	VKM	+	+	+	+
9	*B. pumilus*	VKM B-749	VKM	−	−	+	+
10	*B. pumilus*	VKM B-750	VKM	+	+	+	−
11	*B. pumilus*	VKM B-935	VKM	−	−	+	−
12	*B. pumilus*	VKM B-1554	VKM	+	+	+	−
13	*B. cereus*	VKM B-504^T^	VKM	−	◊	−	◊
14	*B. cereus*	ATCC 4342	ATCC	−		−	
15	*B. thuringiensis*	VKM B-84	VKM	−		−	
16	*B. thuringiensis*	ATCC 35646	ATCC	−		−	
17	*B. weihenstephanensis*	KBAB4	[54]	−		−	
18	*B. flexus*	AVS-02	Laboratory collection	−		−	
19	*B. subtilis*	168 His III	Laboratory collection	−		−	
20	*B. subtilis*	WB 800n	MoBiTec	−		−	
21	*B. megaterium*	MS941	MoBiTec	−		−	
22	*E. faecium*	FS86	Bifiform^®^	−		−	

«+» Clearing zone detected; «−» no clearing zone; «◊»—Not assessed. Source abbreviations: VKM: All-Russian Collection of Microorganisms; ATCC: American Type Culture Collection; MoBiTec: Company MoBiTec GmbH, Germany.

## Data Availability

The annotated complete genomes of Bolokhovo and Novomoskovsk were deposited into GenBank under accession numbers MT514532 and MT422786, respectively.

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
