# Peer review of "Isolation and Characterization of Two Novel Siphoviruses Novomoskovsk and Bolokhovo, Encoding Polysaccharide Depolymerases Active against Bacillus pumilus"

_ijms, 2022, doi:10.3390/ijms232112988_

Round 1
Reviewer 1 Report
This manuscript “Isolation and characterization of two novel Siphoviruses Novomoskovsk and Bolokhovo encoding polysaccharide depolymerases active against Bacillus pumilus” by Skorynina and co-authors reports characterisation of two new double-stranded DNA bacteriophages that belong to the Siphoviridae family (the order Caudovirales). The authors where able to isolate the phages and characterise their genome. The authors have found that these phages represent two new species of the genus Andromedavirus (class Caudoviricetes). Analysis of some depolymerases from Novomoskovsk and Bolokhovo phages suggests that they can be effective potentially useful for biofilm control. It is rather obvious, that this study needs further detailed studies. The manuscript is well written in general and provide essential information related to the characterisation of newly found phages
The only questions that it was not clear for a reader why the authors decided to concentrate on the polysaccharide depolymerases. It did not sound as a very specific issue. More information on that part of the manuscript should be provided.
There are some issues that should be addressed by the authors prior the publication:
It was confusing: the page 3, lines 97-99 the authors write that it was found the phages can infect nine and eleven of 12 B. pumilus strains. In table 1 the authors indicate that the phages could not infect seven other Bacillus strains and E. faecium. Later, in the conclusions, the authors write “Novomoskovsk and Bolokhovo were able to infect most of the B. pumilus strains tested, however, they could not infect seven other Bacillus strains and E. faecium”. It was difficult to find the consistence between the table 1 and the text-> the numbers of pluses in the table 1 did not correspond to the text information. It seems that the authors were not consistent in their numbers, and they make the analysis of the information provided rather difficult by writing some different numbers in the different parts of the manuscript.
It is a bit strange approach when in the figure legends the authors write (Figure 3 and 4) “Functionally assigned ORFs are coloured according to their general functions (see the legend)”, but nothing is given in the figure legend, some colour labelling in shown in the centre of the circle of the genome map, which is not a figure legend.
Figure 5, the most interesting part -> panel a -> it is so small that is really difficult to see anything and the text with numbers is not readable at all. Please make the panel A as two lines: first part form the left side to the middle, and the second part) below the first one) from the middle to the end (put the second one under the first one, the width of these subpanels has to correspond to the width of the page), so everybody will be able to see the relevant information, the text and genomes.
Again, the authors write in the figure legend 5: “the gray regions between genome maps indicate the level of identity (see the legend)”. Nothing is written in the figure legend. The authors should provide the proper explanations, and where the levels of identity can be found.
Text lines 191-194 is rather confusing: the authors write “An HHpred analysis revealed high similarity of both Gp28 and Gp31 to the domains and proteins of the Pectinlyase-like superfamily (SCOPe ID: b.80.1) [46]: probability values of 99.81% and 99.7%, respectively, were obtained in the alignments with rhamnogalacturonase A of Aspergillus aculeatus (SCOPe ID: d1rmga_)”. While Gp28 and Gp31 are so similar (nearly identical to) the proteins from Pectinlyase-like superfamily they were assigned to phage depolymerases. So, one can make a conclusion that these two proteins are rather similar both in structure and function. However, further (lines 201-203) information says something opposite: that the proteins are 29.73% identical to each other (line 199). The authors now write that there is significant divergency between the proteins and they authors make a conclusion that “Gp28 and Gp31 fall into different clades. Thus, it is reasonable to expect the two proteins to differ considerably in their key characteristics, such as affinity to bacterial glycocalyx”. It suddenly became clear that the proteins have different functions: which characteristics were so much different; that was not explained. It seems that more detailed information must be provided, and the authors have to specify, where they found the differences and how it is linked to the differences in their function. This part is not complete and confusing.
The final small comments: Where is a section numbered as 3? There is 2: Results; then: 4 -> Materials and Methods. It is followed by 5 -> Conclusions. What is supposed to be in part 3 which is absent?
Author Response
Dear Reviewer,
Thank you very much for your valuable corrections and suggestions.
The Following changes have been made to the manuscript:
Point 1: It was confusing: the page 3, lines 97-99 the authors write that it was found the phages can infect nine and eleven of 12 B. pumilus strains. In table 1 the authors indicate that the phages could not infect seven other Bacillus strains and E. faecium. Later, in the conclusions, the authors write “Novomoskovsk and Bolokhovo were able to infect most of the B. pumilus strains tested, however, they could not infect seven other Bacillus strains and E. faecium”. It was difficult to find the consistence between the table 1 and the text-> the numbers of pluses in the table 1 did not correspond to the text information. It seems that the authors were not consistent in their numbers, and they make the analysis of the information provided rather difficult by writing some different numbers in the different parts of the manuscript.
Response 1: The discrepancy between the data in Table 1 and the number of sensitive strains provided in the Conclusions has been corrected.
Point 2: It is a bit strange approach when in the figure legends the authors write (Figure 3 and 4) “Functionally assigned ORFs are coloured according to their general functions (see the legend)”, but nothing is given in the figure legend, some colour labelling in shown in the centre of the circle of the genome map, which is not a figure legend.
Response 2: Figure legends have been rewritten in a clearer way (Figures 2, 3 and 4 in the updated version).
Point 3: Figure 5, the most interesting part -> panel a -> it is so small that is really difficult to see anything and the text with numbers is not readable at all. Please make the panel A as two lines: first part form the left side to the middle, and the second part) below the first one) from the middle to the end (put the second one under the first one, the width of these subpanels has to correspond to the width of the page), so everybody will be able to see the relevant information, the text and genomes.
Response 3: We have decided not to split Figure 5a into two parts, but rather scale it up to full width of the page (~170% from previous version) and improve its quality - now if you zoom in on the page, all the details are readable. We hope you consider this a sufficient improvement.
Point 4: Again, the authors write in the figure legend 5: “the gray regions between genome maps indicate the level of identity (see the legend)”. Nothing is written in the figure legend. The authors should provide the proper explanations, and where the levels of identity can be found.
Response 4: The text has been corrected to “The gray regions between the genome maps indicate the level of identity, from low (light-gray) to high (black, see the coloring scheme on the top)”.
Point 5: Text lines 191-194 is rather confusing: the authors write “An HHpred analysis revealed high similarity of both Gp28 and Gp31 to the domains and proteins of the Pectinlyase-like superfamily (SCOPe ID: b.80.1) [46]: probability values of 99.81% and 99.7%, respectively, were obtained in the alignments with rhamnogalacturonase A of Aspergillus aculeatus (SCOPe ID: d1rmga_)”. While Gp28 and Gp31 are so similar (nearly identical to) the proteins from Pectinlyase-like superfamily they were assigned to phage depolymerases. So, one can make a conclusion that these two proteins are rather similar both in structure and function. However, further (lines 201-203) information says something opposite: that the proteins are 29.73% identical to each other (line 199). The authors now write that there is significant divergency between the proteins and they authors make a conclusion that “Gp28 and Gp31 fall into different clades. Thus, it is reasonable to expect the two proteins to differ considerably in their key characteristics, such as affinity to bacterial glycocalyx”. It suddenly became clear that the proteins have different functions: which characteristics were so much different; that was not explained. It seems that more detailed information must be provided, and the authors have to specify, where they found the differences and how it is linked to the differences in their function. This part is not complete and confusing.
Response 5: Regarding lines 191-194 and 201-203:
A high Probability value in HHpred does not guarantee high identity in BlastP, since the tools employ different algorithms. Unlike BlastP, based on the pairwise comparison of individual sequences, HHpred compares profile hidden Markov models (HMMs), and is therefore a much more sensitive tool able to establish connections to remotely homologous proteins.
HHpred takes into account secondary structure similarity when calculating the HHpred Probability value for each hit. Along with a Probability value, for each alignment HHpred also calculates an Identity value in a way similar to how it is done in BlastP, with the difference that it is profiles that are compared in HHpred, not individual sequences. Since the Probability and Identity are different parameters, it is not uncommon to obtain a high Probability value in an HHpred alignment where the Identity value is low. Citing the creators of HHpred:
‘The probability value reported by HHpred for a match to be a true positive is the most important criterion to infer if a match is homologous to the query or is just a high-scoring chance hit. When it is greater than 95%, evolutionary relatedness is highly likely.’ [Gabler, Felix, et al. "Protein sequence analysis using the MPI bioinformatics toolkit." Current Protocols in Bioinformatics 72.1 (2020): e108.]
In the HHpred alignment of Gp28 with the rhamnogalacturonase A of Aspergillus aculeatus, the Probability value was 99%, while Identity was only 13%. The same Probability value and a similar Identity value were obtained in the alignment of Gp31 with the same rhamnogalacturonase A.
In the Manuscript, we provide the HHpred Probability values for both alignments (99%), and also the Identity value obtained in a BlastP alignment of Gp28 and Gp31 (29.73%) - in order to illustrate that even though the two proteins are divergent in terms of their primary structures (low BlastP identity), they are, most likely, functionally homologous (high HHpred Probability, meaning highly similar predicted secondary structures).
Point 6:
The final small comments: Where is a section numbered as 3? There is 2: Results; then: 4 -> Materials and Methods. It is followed by 5 -> Conclusions. What is supposed to be in part 3 which is absent?
Response 6: We decided to write one section "Results and Discussion" instead of splitting it into two sections. Section numbering corrected. The title of section 2 has been corrected.
Reviewer 2 Report
The authors present a structured study of two new phages infecting members of the Bacillus pumilus group and highlight the potential future use of the phage-derived depolymerases. The careful performance of the analyses seem to be useful for further studies of the presented phages. However, the authors did not include a discussion or result/discussion section, which makes it impossible to evaluate this manuscript for publication. In addition, for a thorough characterization of the phages, the authors should add an experimental approach to study the receptors of the presented phages and their one-step growth curves and adsorption times. This might also be discussed if not conducted. Phage-derived depolymerases as antibacterial agents are becoming increasingly important in light of the antibiotic resistance crisis. The depolymerases in this study could be further analyzed by size exclusion chromatography, spectroscopy, stability analysis, etc. At the very least, these steps should be discussed.
Author Response
Dear Reviewer,
Thank you very much for your valuable corrections and suggestions.
The Following changes have been made to the manuscript:
Point 1: However, the authors did not include a discussion or result/discussion section, which makes it impossible to evaluate this manuscript for publication.
Response 1: We decided to write one section "Results and Discussion" instead of splitting it into two sections. Section numbering corrected. The title of section 2 has been corrected.
Point 2: In addition, for a thorough characterization of the phages, the authors should add an experimental approach to study the receptors of the presented phages and their one-step growth curves and adsorption times. This might also be discussed if not conducted.
Response 2: A one-step growth assay and an adsorption assay have been conducted (Sections 2.5, 3.7 and 3.8). Also, Time-kill Assay of Bolokhovo and Novomoskovsk Phages placed as a separate section and supplementary figure S3 transferred to the main text of the manuscript (Section 2.4).
Point 3: Phage-derived depolymerases as antibacterial agents are becoming increasingly important in light of the antibiotic resistance crisis. The depolymerases in this study could be further analyzed by size exclusion chromatography, spectroscopy, stability analysis, etc. At the very least, these steps should be discussed.
Response 3: The recombinant depolymerases have been analyzed by size exclusion chromatography. The results are discussed in Section 2.8, lines (260-266). Also, stability of the depolymerases has been evaluated in temperatures ranging from 4 to 80°C and pH ranging from 4.0 to 12.0 (Sections 2.10, 3.14), and strain specificity of the enzymes has been determined (Sections 2.9, 3.13, Table 1).
Reviewer 3 Report
The manuscript describes and discusses logically designed experiments and presents results that are expected to be of large interest for the scientific community. It is an interesting study with an interesting approach. The paper in the whole is well designed and results sound. Nevertheless, the manuscript needs a minor revision:
Point 1: In the introduction part should be more highlighted the main aim of the paper, and additionally, what is the novelty of carried research work.
Point 2: How do the Authors select the analytes? The rational of the choice of the selected biologically active compounds studied is missing and should be clearly discussed.
Point 3: Quality of the figures must be improved.
Author Response
Dear Reviewer,
Thank you very much for evaluating our manuscript and for your valuable corrections and suggestions.
Point 1: In the introduction part should be more highlighted the main aim of the paper, and additionally, what is the novelty of carried research work.
Response 1: The Introduction has been updated in order to highlight the main focus of the study (lines 84-86) and explain the scientific rationale behind this work (lines 63-68) and the novelty of the research performed (lines 82-84). The quality of Figures 4, 5, 6 and 8 has been improved, now all the details can be seen clearly.
Point 2: How do the Authors select the analytes? The rational of the choice of the selected biologically active compounds studied is missing and should be clearly discussed.
Response 2: The rational of choice of B. pumilus phages included into introduction section. More experiments were conducted (as requested by Reviewer2) and results and its discussion included Results and Discussion Section.
Point 3: Quality of the figures must be improved.
Response 3: The figures qualities was improved.
Round 2
Reviewer 2 Report
Suggestion to Editors: Accept